# The Social Solidarity Economy and the Hull-House Tradition of Social Work: Keys for Unlocking the Potential of Social Work for Sustainable Social Development

Michael Emru Tadesse * and Susanne Elsen

ASTRA Project, Faculty of Education, Free University of Bozen-Bolzano, Regensburger Allee 16—viale Ratisbona 16, 39042 Bressanone-Brixen, Italy; susanne.elsen@unibz.it
* Correspondence: michaelemru.tadesse@unibz.it

**Abstract:** Social work (SW) is criticized for its (i) inconsistent ontology, epistemology, and methodology and (ii) co-dependency with the capitalist hegemony, which is the main cause of multiple crises that primarily affect the most vulnerable. Addressing these issues is of paramount importance if SW is to achieve its core mandate of promoting social change, social development, social cohesion, and the empowerment and liberation of people. The purpose of this paper is to assert that SW can address these issues by learning from the (i) Settlement House Movement (SHM), especially the Hull-House tradition of SW; and (ii) current endeavors of the Social Solidarity Economy (SSE). We were led to this assertion because we noticed in our research, in the area of SSE of vulnerable groups and SW, remarkable similarities and potentials of these two approaches to help transform SW. Our argument is based on data and insight gained from (i) a narrative literature review on the history of SW and the nature of SSE; and (ii) a systematic scoping review of the SSE of People of African Descent (PAD) in Europe. In this paper, we elaborate on our key arguments and provide examples and recommendations.

**Keywords:** Jane Addams; Hull-House; migration; people of African descent; Settlement House Movement; social development; social sustainability; social solidarity economy; social work

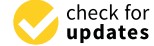



## 1. Introduction

Social Work (SW) is known as the "helping profession" ([NASW n.d.a](#)). While it is commonly associated with helping individuals and families in immediate situations of crisis, it could also be considered as the primary profession in the case of social sustainability as it addresses various social problems with implications for social, economic, and environmental sustainability. The current global definition of SW shows that SW is more than merely helping vulnerable individuals and families. "[SW] is a practice-based profession and an academic discipline that promotes social change and development, social cohesion, and the empowerment and liberation of people. Principles of social justice, human rights, collective responsibility and respect for diversities are central to [SW]" ([IFSW n.d.](#), para. 2).[1]

However, SW has been criticized for not facing up to its two interrelated internal inconsistencies/contradictions that hinder it from realizing its mission and mandate in a politically accountable manner. These contradictions are the (i) co-dependency between SW and the capitalist hegemony and (ii) lack of consistency in SW's ontological, epistemological, and methodological dimensions ([Bell 2012](#); [Boetto 2017](#); [Coates 2003](#)).

In the former case, SW is criticized for its role in sustaining the root cause of the very social problems it promises to address. SW is sustaining the capitalist hegemony that has created and exacerbated the multiple social, economic, and ecological crises that primarily affect the most vulnerable individuals, groups, communities, and societies. Such criticism is best evidenced in the mutual codependency between SW, the welfare state (which hosts SW), and industrial capitalism (which hosts both SW and the welfare state) ([Boetto 2017](#);

Coates 2003). To find employment, resources, and a client base, social workers rely on human service organizations, which are part and parcel of the welfare state of the capitalist hegemony that uses neoliberal and managerialist notions of welfare (Healy 2014). While working in such a context, social workers can be seen as contributing to sustaining the capitalist hegemony by helping people to adapt to and participate in an economy and society characterized by neoliberal values of individualism and competition (Boetto 2017; Coates 2003).

In the latter case, the inconsistency is between SW's (i) dominant modernist and positivist root/ontology and methodology and (ii) epistemological base and contemporary mission (Boetto 2017). Mainstream SW predominantly depends on paradigms that are based on modernist and positivist ontology, resulting in the use of biomedical (Bell 2012) and psychotherapy-based models of intervention (Specht and Courtney 1995).

Ontology can be defined as ways of being or a worldview about existence or reality (Parton and O'Byrne 2000). Various authors argue that the dominant ontological base of SW is modernist, underpinned by individualism, rationalism, universalism, and neoliberal ideology (Bell 2012; Boetto 2017). This ontological base creates a rational–technical and positivist approach to SW and "reinforces oppressive hierarchies" (Bell 2012). While the above argument is true in terms of identifying the dominant ontology in SW, we would like to remind readers that SW has had at least two competing traditions, i.e., (i) the aforementioned dominant positivist and individualist path followed by scholars such as Mary Richmond; and (ii) the largely ignored pragmatist and community-based tradition followed by scholars such as Jane Addams, who is considered one of the main radical pragmatist public philosophers (Hamington 2022; Hamington n.d.; Morrison 2016).

Epistemology is defined as ways of knowing (Parton and O'Byrne 2000) or the thinking dimension of practice. SW's epistemological base is primarily about its (i) professional values, principles, and ethical standards; and (ii) professional knowledge (Boetto 2017). SW's professional values, principles, and ethical standards are about the inherent worth and dignity of human beings, human rights, social justice, collective responsibility, and respect for diversity. SW's professional knowledge is both interdisciplinary and transdisciplinary, drawing on a wide array of theories and research from, among others, SW itself, community development, social pedagogy, sociology, anthropology, ecology, economics, education, psychology, and public health (IFSW n.d.). SW's epistemology contradicts and goes beyond its dominant modernist ontology (Boetto 2017). Unlike its dominant ontology and methodology, SW's epistemological bases (especially its values, principles, and ethical standards) are in line with its true mission/mandate, i.e., promoting social change and development, social cohesion, and the empowerment and liberation of people (IFSW n.d.).

Methodology refers to the doing (actions, interventions, practice) aspects or strategies of SW (Boetto 2017). The dominant methodology of SW, instead of being informed by its epistemological base that features a collectivist/cooperativist nature, has largely been informed by its dominant modernist tradition that emphasizes individualist strategies such as individual and family casework. As a result, transformative practices such as community work have been marginalized, especially in the West (Bell 2012; Boetto 2017). This shows that there is a need in SW for creating consistency among its ontological, epistemological, and methodological foundations.

The two major inconsistencies discussed above have made SW less effective at tackling the social problems it strives to eradicate. To contribute toward addressing such inconsistencies and making SW a more holistic and impactful profession, we propose in this paper that SW transform itself by (i) looking back to its true but largely neglected tradition of SHM, epitomized by the works of Jane Addams and her colleagues at Hull-House; and (ii) examining current endeavors in the field of Social Solidarity Economy (SSE) that are contributing significantly toward the achievement of SW's mission.

## 2. Methods

To support our argument, we discuss three points. Firstly, we discuss the history of SW, emphasizing Jane Addams' and Hull-House's work in relation to the condition of European immigrants in the USA. To do so, we examined the origin of SW as a profession by looking at historical records (books, articles, and other materials) written by prominent SW scholars including Addams. For instance, we collected evidence from the book by Addams titled *Twenty Years at Hull-House: With Autobiographical Notes*. We also visited the websites and publications of national and international SW associations such as the American National Association of Social Workers (NASAW) and the International Federation of Social Workers (IFSW).

Secondly, we discuss the concept and examples of SSE. To do so, in addition to a narrative literature review on the SSE, we used the results and insights of a research project we are carrying out on the SSE of vulnerable local communities in Europe. We particularly used our systematic scoping review on the SSE organizations (SSEOs) of People of African Descent (PAD) in Europe.[2] We emphasized PAD in Europe as an example to show the condition of current immigrants. Figure 1 below shows the process of identification and selection of articles included in the scoping review.

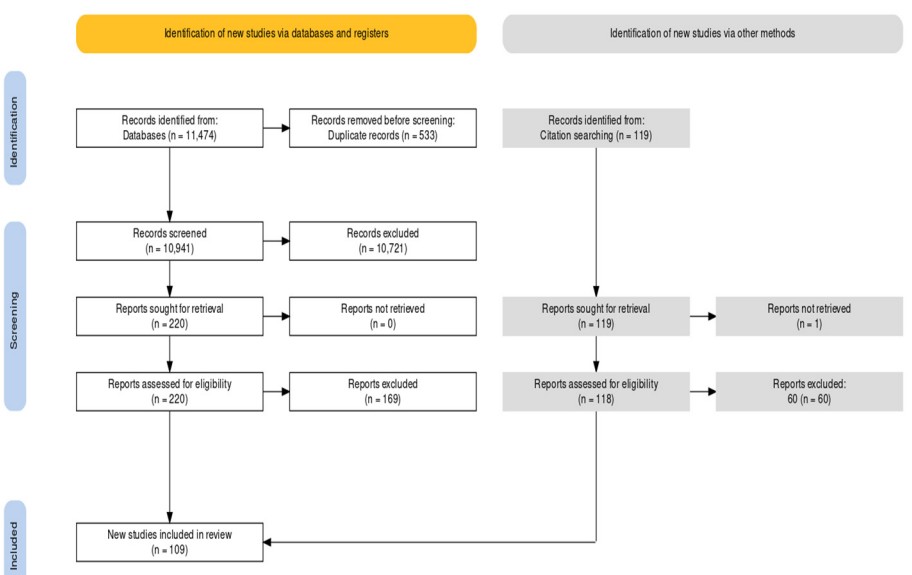

**Figure 1.** PRISMA flow diagram (Haddaway et al. 2022).

Finally, based on the information gained from the above two points, we show the parallel between the Hull-House tradition and SSE.

## 3. Results and Discussion

### 3.1. The History of Social Work and Its Two Competing Traditions

3.1.1. Motivating Factors for the Emergence of SW as a Profession

SW is a profession which originated in the USA in the late 19th century in response to a myriad of social problems including the economic and social poverty created as a result of the Industrial Revolution and modernism (logical–positivist rationality). These problems were particularly associated with the then mass migration "crises" involving European immigrants in the USA (Franklin 1986; NASW n.d.b).

Between 1850 and 1913, over 27 million people emigrated to the USA from Europe. In the early stage, immigrants from northwest Europe were the majority. However, after 1890, the so-called "new emigrants" emigrating from Southern Europe (Italy, Portugal, and Spain) and Eastern Europe (Austria, Hungary, and Poland) became prevalent. Most of these immigrants were unskilled, young, single men who were predominantly economic

immigrants (Hatton and Williamson 1994). These people emigrated to the USA for various reasons. Many emigrated because they had experienced crop failure, land and job shortages, rising taxes, famine, etc., and perceived the USA as the land of opportunity. Many others emigrated in search of freedom from political, religious, and other persecution (Library of Congress n.d.).

In the USA, the Industrial Revolution caused enormous changes in society. One such change was associated with a labor shortage and the need for an enormous number of both skilled and unskilled laborers to increase industrial production. The newly arrived immigrants from different European countries contributed to filling this demand for laborers. However, poor living and working conditions and low wages, coupled with the prevailing ideologies of that time (such as social Darwinism, Calvinism, and Liberalism, which emphasized moral certainty in which social problems were perceived as individuals' problems and personal responsibilities), created and exacerbated the vulnerability of the immigrants. Immigrant slums, characterized by delinquency, disease, and poverty, rapidly grew along with the industrial cities (Franklin 1986).

Furthermore, anti-immigrant sentiment (e.g., racism, antisemitism, and xenophobia) against Southern and Eastern Europeans, Asians, and African Americans was held by many intellectuals and white Americans who believed in the theories of social Darwinism and eugenics which became prevalent in the late 19th and early 20th centuries, exacerbating the situation (Hirschman 2006).

To deal with such problems, various early founders of SW devised and emphasized different approaches, resulting in two traditions of SW. These traditions are associated with two movements in the USA, i.e., the Charity Organization Society (COS) Movement and the Settlement House Movement (SHM), both of which came into existence at the same time. In other words, the beginning of professional SW can be associated with the works of the American COS and SHM (mainly Hull-House). The movements have two things in common. The first is that both adapted their model from corresponding movements in England. The second is that both were interested in scientific SW (Franklin 1986). Other than that, these two traditions have been competing for many decades, creating ideological, theoretical, and methodological tensions that persisted within SW between the individualized treatment camp (i.e., the COS tradition) and the social reform camp (the SHM tradition) (Franklin 1986; Thompson et al. 2019).[3]

3.1.2. American Charity Organization Societies and Mary Richmond (1861–1928)

The first American COSs attempted to address social problems with "scientific charity" starting in 1877 (Simmons University n.d.), creating a tradition that focuses on individualized SW.

This tradition of SW was advocated mainly by those who believed in the philosophy of liberalism, underscoring the centrality of individual responsibility and action. The poor were considered lazy or morally deficient. This tradition aimed to make almsgiving scientific, efficient, and preventative by assessing who can be considered as the deserving poor and rehabilitating them without addressing environmental or structural constraints. To do so, this tradition relied on case conferences and "friendly visiting" (Franklin 1986), first by volunteers known as friendly visitors (in the 1880s) and then by paid employees (in the 1890s) (Lubove 1968).

This tradition has had prominent advocates and pioneers such as Mary Richmond (1861–1928), who was a career COS administrator and one of the most influential figures in SW (Franklin 1986; Stuart 2019). In addition to helping transform the organizational aspects of COS organizations, Richmond is known for establishing the conceptual base of social casework for working with individuals and families. Her casework methods were published as a book in 1917 with the title *Social Diagnosis* (Stuart 2019). Her work was heavily influenced by people and concepts from the biological sciences and medical professions. Furthermore, her work was influenced by people who believed in moral certainty and social Darwinism (Franklin 1986).

This tradition promoted individualized practice or the rehabilitation of individuals in terms of individual and family casework (Franklin 1986). Based on this, it provided SW training and education. In 1898, the COS established the first summer school in SW at Columbia University in New York City (NASW n.d.b). This was followed by a full-time graduate program at the New York School of Philanthropy in 1904 (Columbia School of Social Work n.d.).

However, most of COS's scientific charity relief efforts were found to be ineffective. One reason for this was that many recipients favored mutual aid schemes organized by their communities (Simmons University n.d.). Despite its ineffectiveness, this tradition remained dominant in SW in terms of maintaining individualistic SW. This influence is seen especially in relation to the development of psychiatric and psychotherapy-based SW after, 1919 (in the USA) as well as the introduction of managerialism in SW (in Europe) in the 1970s. Psychiatric and psychotherapy-based SW went even further in its individualistic focus, losing the social aspect of its SW practice. In their book titled *Unfaithful Angels: How Social Work Has Abandoned Its Mission*, Specht and Courtney (1995) sharply criticized the development and dominance of psychiatric and psychotherapy-based SW in the USA as something that goes far beyond the COS's tradition of helping poor people and addressing social problems.

Psychiatric and psychotherapy-based SW was later accompanied by managerialism in SW, which imported notions of quality assurance from the manufacturing industry of the neoliberal capitalist system (Adams 1998).[4] Managerialism in SW is informed by neoliberal ideology's assumption that the market is superior to other sectors and therefore social services need to be managed like the private sector while focusing on the reduction of expenditure and the expansion of "consumer" choice. Accordingly, the success of social workers was measured in terms of managers' targets or organizational needs instead of service users' needs. This change has resulted in the "deformation" of the profession by making SW forget its mission of addressing social problems using either direct work with individuals, families, groups, and communities or indirect works such as advocacy and social policy (Rogowski 2011).

3.1.3. Settlement House Movement and Jane Addams (1860–1935)

The second tradition of SW is the SHM, especially that of Hull Settlement House. This tradition had prominent advocates including Jane Addams (1860–1935), who is one of the most influential and famous figures in SW (Franklin 1986). Addams is known as the mother of SW, at least in the USA.[5] Furthermore, she is described as a public philosopher, social reformer, social politician, social researcher, community organizer, peace activist, and founder of emancipatory SW (Elsen 2011). She is also the receiver of the 1931 Nobel Peace Prize for her contributions to just and peaceful societies (The Nobel Peace Prize 1931).

Inspired by her visit to London's Toynbee Hall (which was the first settlement house founded in 1884),[6] Addams, along with Ellen Gates Starr, established the Hull-House in 1889 in Chicago to address the myriad of social problems affecting the poor, such as working-class European immigrants (Addams 2008; Eberhart 2009; NASW Foundation n.d.).

In the beginning, Addams and Starr rented an abandoned residence built by Charles G. Hull in 1856 at 800 South Halsted Street in Chicago (Addams 2008; Encyclopaedia Britannica n.d.a). They then expanded Hull-House into a 13-building complex covering an entire city block and including facilities such as a gymnasium, theater, art gallery, libraries, pools, classrooms, a kindergarten, dormitories, etc. (Addams 2008; National Park Service n.d.). Addams described Halsted Street and the condition of the immigrants as follows:

> Hull-House once stood in the suburbs, but the city has steadily grown up around it and its site now has corners on three or four foreign colonies. . . . The policy of the public authorities of never taking an initiative, and always waiting to be urged to do their duty, is obviously fatal in a neighborhood where there is little initiative among the citizens. The idea underlying our self-government breaks down in such a ward. The streets are inexpressibly dirty, the number of schools inadequate,

sanitary legislation unenforced, the street lighting bad, the paving miserable and altogether lacking in the alleys and smaller streets, and the stables foul beyond description. Hundreds of houses are unconnected with the street sewer. The older and richer inhabitants seem anxious to move away as rapidly as they can afford it. They make room for newly arrived immigrants who are densely ignorant of civic duties. This substitution of the older inhabitants is accomplished industrially also, in the south and east quarters of the ward. The Jews and Italians do the finishing for the great clothing manufacturers, formerly done by Americans, Irish and Germans, who refused to submit to the extremely low prices to which the sweating system has reduced their successors. (Addams 2008, chp. V)

By establishing the Hull-House, where they lived and worked in an impoverished neighborhood with the poor,[7] and by introducing experimental and innovative approaches to dealing with social problems, Addams and her colleagues fought for social justice and social change (Addams 2008; Melgar et al. 2021). Their approach was different from the COS tradition. They believed in pragmatism and acknowledged the social causes of individuals' poverty. They located the main cause of individuals' poverty in the macrosystem or social conditions. Accordingly, they promoted collaborative group and community-based SW, socialization, and social action/reform (Addams 2008; Franklin 1986; Melgar et al. 2021). In this case, their activities and services were directed toward all people who lived in the community, not just people who had problems (Pumphrey and Pumphrey 1961; Thompson et al. 2019). This means their services were designed to improve living and working conditions and deal with the sources of social problems (Trattner 2007). Furthermore, they believed that respect for the dignity of all people was fundamental (Melgar et al. 2021) and that receiving social support should not be a degrading experience (NASW Foundation n.d.).[8]

The Hull-House tradition was like a university branch for the development of sociocultural, political, and economic innovations in disadvantaged communities. It was also the origin and local crystallization point of a comprehensive pragmatic SW. Emancipatory educational and sociocultural work, in connection with social, labor market, and economic policy reforms, defined its work. In its work, cooperative self-help and SSE had central roles (Elsen 2011).

The Hull-House itself was a community center and an experiment in collective living (Addams 2008; Eberhart 2009). The Hull-House involved various community development initiatives that started immediately after its establishment. For example, within four months of its establishment, there were 13 clubs and 38 different courses per week at Hull-House (Eberhart 2009).

Throughout its 120 years of existence (1889–2012), the Hull-House engaged in diverse group- and community-oriented initiatives addressing various needs of the community. These included clubs, cooperatives, an art gallery, a music school, concerts, a theater, a labor museum, scientific projects, public schools, a day nursery and kindergarten, workshops and vocational training (e.g., wood, metal, and electrical work), college and university extension courses and summer schools, public discussions, a reading-room and public library, a coffeehouse and public kitchen, a gymnasium, a public bathroom, a public playground, a paving of streets project, garbage removal and public sanitation initiatives, public health measures and a clinic, a free employment bureau, trade unions, and federations (e.g., the Chicago Federation of Settlements and the National Federation of Settlements) (Addams 2008; Eberhart 2009; VCU Libraries Social Welfare History Project n.d.). To provide further examples, we will briefly discuss some of these group- and community-oriented initiatives and the contribution of Hull-House in terms of addressing community needs.

***Kindergartens and Day Nursery.*** The kindergartens and day nursery were Hull-House's first organized undertakings, aiming to help working mothers and their children. There were at least two kindergartens at Hull-House. The nursery was able to host 30 children per day on average. The nursery was run by Hull-House for 16 years before it was handed over to the United Charities of Chicago (Addams 2008; The Working Centre n.d.).[9]

*Working People's Social Science Club.* Another early undertaking of Hull-House was the Working People's Social Science Club. The club provided a platform for the discussion of various socioeconomic subjects openly and democratically. Members of the club met weekly to hear speeches about a particular topic and participate in lively debates and discussions. The meetings were attended by 40–100 people and existed for 7 years (Addams 2008; VCU Libraries Social Welfare History Project n.d.).

*The Hull-House Women's Club*. This club had around 600 members and a building of its own. The club hosted, among others, lectures on current topics by distinguished speakers, discussions by club members, and musical afternoons led by the club's own chorus. The club also had a visiting nurse who lived at Hull-House (The Working Centre n.d.).

*The Hull-House Boys' Club.* The club had around 1500 members and occupied its own building with bowling alleys, billiard tables, athletic apparatus, a library, classrooms, and shops for work in iron, wood, and printing (The Working Centre n.d.).

*The Hull-House Men's Club.* This club was established in 1893 and contributed to the early attempts of Hull-House in relation to civic cooperation. Among other things, the club engaged in political struggles and held occasional public debates (Addams 2008; The Working Centre n.d.).

*Public Baths*. Hull-House offered shower baths in its basements for the use of the neighborhood. By doing so, Hull-House provided crucial experience and an argument (convincing public authorities) for the establishment of the first public bathhouse in Chicago (Addams 2008; The Working Centre n.d.).

*A Public Kitchen and Coffee-House.* Hull-House cooperatively ran a public kitchen (On the Commons 2010). The kitchen was opened to address problems identified by several studies regarding the sweatshops in the neighborhood and the dietary conditions of various immigrants. The results of the investigation showed that immigrants and sewing women, because of their working conditions and inadequate pay, were unable to properly feed their families. The public kitchen made available properly cooked and nutritious food, even though it was not popular among immigrants coming from different countries with different tastes. However, the coffeehouse became a popular and self-sustaining social center of the neighborhood (Addams 2008; The Working Centre n.d.).

*The Hull-House Labor Museum.* The Labor Museum was created to help older generations of immigrants coming from different countries to show and teach their preindustrial craft practices (e.g., in textile manufacturing, metal working, and bookbinding) that they brought from their countries of origin to the second and third generations of immigrants. This initiative helped to connect workers in the city's factories and sweatshops. It also made children have greater respect for their parents. Furthermore, the sale of handcrafted products created economic opportunities for the immigrants (Addams 2008; The Jane Addams Papers Project n.d.).

*The Chicago Arts and Crafts Society.* This Society was closely associated with the Labor Museum and classes in pottery, metalwork, enamel, and woodcarving. Successful classes in drawing, modeling, painting, and lithography were carried out in addition to art exhibits (Addams 2008; The Working Centre n.d.).

*Employment Bureau.* Hull-House strived to procure free employment bureaus to deal with unemployment and related problems until a law authorizing such bureaus was passed in 1899. It maintained its informal work in securing employment for immigrants and facilitated various meetings to discuss the problems of the unemployed. It also helped in securing the enforcement of the employment agency law (Addams 2008; VCU Libraries Social Welfare History Project n.d.).

*Cooperatives.*[10] Despite its significance, the effect and involvement of Addams and Hull-House in cooperatives is the least discussed topic (Eberhart 2009). Addams had immense interest in cooperatives and believed in the "persistency of the cooperative ideal" even in individualistic countries such as the USA. She hosted cooperative congresses, exchanged ideas with cooperators in Chicago, studied many successful and unsuccessful cooperative experiments in different places, and helped to organize several cooperatives at Hull-House (Addams 2008).

The first such cooperative was the Housing Cooperative for Young Working Women, also known as Jane Club, which was initially established to protect striking working girls from capitulation (to help them stand by one another). This cooperative was a success story (Addams 2008). Addams described the establishment of the cooperative as follows:

> We read aloud together Beatrice Potter's little book on 'Cooperation,' and discussed all the difficulties and fascinations of such an undertaking, and on the first of May, 1891, two comfortable apartments near Hull-House were rented and furnished. The Settlement was responsible for the furniture and paid the first month's rent, but beyond that, the members managed the club themselves. The undertaking 'marched,' as the French say, from the very first, and always on its own feet. Although there were difficulties, none of them proved insurmountable, which was a matter for great satisfaction in the face of a statement made by the head of the United States Department of Labor, who, on a visit to the club when it was but two years old, said that his department had investigated many cooperative undertakings, and that none founded and managed by women had ever succeeded. At the end of the third year the club occupied all of the six apartments which the original building contained, and numbered fifty members. . . . In the course of time a new club house was built by an old friend of Hull-House much interested in working girls, and this has been occupied for twelve years by the very successful cooperating Jane Club. (Addams 2008, chp. VII)

This cooperative was self-governing from the beginning. The members also made a weekly contribution of USD 3, with an occasional small assessment, which enabled them to meet their expenses in relation to rent, food, heating, and other services (The Working Centre n.d.).

The second such cooperative was the Hull-House Cooperative Coal Association, which was established in 1892 to improve the economic conditions of members by making individual efforts more effective through an organization. The association developed a large membership and sustained itself for three years before the experiment was terminated by the cooperators because of too much emphasis on its philanthropic side, which clashed with the business aspect (Addams 2008). The cooperative provided low-priced coal to 19th ward residents of Chicago during cold winters (On the Commons 2010).

There were also two other cooperative clubs for young men, i.e., the Phalanx Club, which existed from 1892 to 1895, and the Culver Club, which existed from 1907 to 1910 (VCU Libraries Social Welfare History Project n.d.). The Phalanx Club, like the Jane Club, implemented the same plan of cooperative living, except for kitchen-related activities, on another two-story frame building (Moore 1897). The Culver Club was also a self-sustaining housing cooperative of 30 working boys who occupied two upper floors of the Hull-House Boys' Club Building. The club contributed positively to the social life of the Boys' Club House (The Working Centre n.d.).

*Trade Unions.* Hull-House contributed to the formation and development of trade unions in Chicago. This was possible partly because organizers of various women's unions lived as residents at Hull-House. Several unions were established at Hull-House. For example, the Chicago branches of two federal organizations of working women, i.e., the Women's Union Label League and the Women's Trade-Union League, were established at Hull-House. Furthermore, Hull-House was a place where several unions held their regular meetings (Addams 2008; The Working Centre n.d.).

*Social Movements*. The Hull-House also contributed to various progressive social movements and policy initiatives such as the peace movements (e.g., anti-WWI movement), women's suffrage movement, proletarian women's movement, and child rights movement (e.g., the Federal Child Labor Law) (Addams 2008). Regarding their first lobbying experience, Addams wrote: "The Hull-House residents that winter had their first experience in lobbying. I remember that I very much disliked the word and still more the prospect of the lobbying itself, and we insisted that well-known Chicago women should accompany this first little group of Settlement folk who with trade-unionists moved upon the state capitol in behalf of factory legislation" (Addams 2008, chp. X).

*Social work Education and social research.* Addams and her colleagues at Hull-House and in the SHM significantly contributed to social work and social science education, training, and research. For example, the leaders of the SHM such as Graham Taylor founded the Chicago School of Civics and Philanthropy in 1908 to combine SW education with practice. Instead of using Richmond's curriculum recommendations, they developed a curriculum that emphasizes social theory and philosophy, social reform, and social policy (Franklin 1986; University of Chicago Library 2010). Hull-House residents such as Sophonisba Breckinridge, Grace and Edith Abbott, and Julia Lathrop also contributed to the School and were among the early faculty members, along with George Herbert Mead, Charles R. Henderson, and Ernst Freund (University of Chicago Library 2010).

Addams and Hull-House also had a close collaboration with John Dewey, who was a prominent pragmatist philosopher, psychologist, and educational reformer. For instance, Dewey assigned Addams's books as required reading in his philosophy classes and regularly invited Addams to teach on his courses. Likewise, he was invited, like other university teachers and students, to give lectures to the Hull-House residents. Furthermore, he served on the Hull-House board of directors (Ralston 2022).

In the case of social research, Hull-House and its neighborhoods conducted many pragmatic and reflexive scientific studies ("inquiry as perplexity") that aimed to understand and address various social problems (Elsen 2011). These include: *Investigation of the Sweating System (1892)*; *The Slums of Great Cities (Chicago) (1893)*; *Dietary Investigation*; *Studies in Ward and City Conditions (1895)*; *Investigation of the Saloons of the Nineteenth Ward (1896)*; *Investigation of the Dietary of the Italian Colony (1897)*; *General Study of, 19th Ward*; *Study of Casual Labor on the Lakes (1903)*; *An Intensive Study of the Causes of Truancy (1905)*; *Study of Tuberculosis in Chicago*; *Investigation into the Selling of Cocaine (1907)*; *Study of Midwifery and Study of the Greeks in Chicago (1908)*; *Study of Infantile Mortality among Selected Immigrant Groups (1909)*; and *Investigation of the Home Reading of Public School Children (1910)* (VCU Libraries Social Welfare History Project n.d.). Addams and her colleagues also published the results of their studies and systematic observations. For instance, the residents of Hull-House published some results of their studies in a book titled *Hull-House Maps and Papers* in 1895 (Addams 2008).

In general, the above discussion on the Hull-House tradition provides insights into how SW can meet fundamental human needs and address social problems in a holistic and collaborative manner. However, the Hull-House tradition of SW is not the only one that does so. As discussed in the next section, SSE provides us with similar but contemporary insights.

### 3.2. The Concept and Practice of the SSE

In this section, we discuss how current SSE endeavors, including the SSE of immigrant communities, that are usually practiced outside of the profession of SW, help to achieve the mission of SW by providing various social services and protections.

### 3.2.1. What Is SSE?

SSE can be understood as a sector of the economy (different from the public and private sectors), movement, and theoretical perspective. SSE can generally be defined as a "concept designating enterprises and organizations, in particular cooperatives, mutual benefit societies, associations, foundations and social enterprises, which have the specific feature of producing goods, services and knowledge while pursuing both economic and social aims and fostering solidarity" (ILO 2009, par. 2). As shown in Table 1, the SSE differs from the public and private sectors in several aspects.

SSE assumes that the economy is (a) an open and interdependent subsystem of the ecosystem/environment; and (b) a social construct that can be shaped and reshaped. At the heart of the SSE is the vision of re-embedding economic activities into social, cultural, and ecological contexts to make the economy controlled by local people and to make the economy life-serving instead of life-threatening (Elsen 2018).

**Table 1.** Ideal-type construction of the three sectors of the economy.

|  | **Public** | **Private** | **SSE** |
|---|---|---|---|
| Dominant actors | State | Market | Community |
| Rationality | Distributive | Competitive | Cooperative |
| Response to organizational decline | Voice | Exit | Loyalty |
| Relationship based on governance principle | Hierarchy Control/Dirigisme | Exchange Freedom/Laissez-faire | Solidarity/ReciprocityParticipation |
| Value creation | Public goods | Wealth creation | Blended values (social, ecological, moral, and economic) |

Source: Adopted from Dash (2016, p. 6).

SSE is not a new phenomenon. However, it has not been given enough attention by mainstream economists, SW professionals, politicians, and policymakers. Even scholars who are proponents of SSE have been marginalized for a long time. However, nowadays, we are observing a rapidly growing interest in SSE and other ecosocial innovations as potential solutions to the multiple and intensifying ecological, social, and economic crises that resulted in unsustainability.

SSE contributes to sustainability transformation since it emphasizes social, economic, and, in many cases, environmental objectives (Elsen 2018). Specifically speaking, SSE contributes to the implementation of the 2030 agenda for sustainable development and Sustainable Development Goals (SDGs). Some of the SDGs addressed by SSE include SDG 1—no poverty, SDG 5—gender equality, SDG 8—decent work and economic growth, SDG 9—industry, innovation, and infrastructure, SDG 10—reduced inequalities, SDG 12—responsible consumption and production, SDG 16—peace, justice, and strong institutions, and SDG 17—partnership for the goals (UNRISD 2016).

Furthermore, SSE helps to promote the participation, self-determination, empowerment, and prosperity of social service users. SSE has emancipative power (Elsen 2023). SSE's contribution is especially important when it comes to the social inclusion and empowerment of marginalized and vulnerable communities such as PAD (Hossein 2017).

3.2.2. SSE and Social Services

In welfare states, SSE became more common at the end of the 20th century as a response to growing poverty, unemployment, social exclusion, inequality, austerity/welfare cuts, and other related problems. While most of the responses were bottom-up citizen/community initiatives, some were part of active labor market initiatives or integrative social policy strategies in disadvantaged communities. Among such SSE initiatives, the most successful in the field of social service had the following features: multi-stakeholder structures composed of private and public actors; embeddedness in local communities and networks such as consumer groups; a mix of paid work and voluntary engagement; a mix of own earnings with public and private support; and reinvestment of surplus in the own organization (Elsen 2023).

SSE is where many different types of innovative social services and protections are being provided. What makes the SSE approach different from public services (characterized by state technocratic rationality) and market providers (characterized by profit maximization) is that SSE provides bottom-linked, integrative, and participative context for the development and management of solutions to satisfy human needs and address social problems based on the concrete needs of citizens. SSE is able to do so because it is based on self-organization, collective action, and active citizenship as well as democratic governance of citizens who are affected by a common concern, embedded in local contexts (Elsen 2023). The local level is where social innovations usually occur (UNRISD 2016).

SSE is known for providing social services where people in need actively and meaningfully participate as co-producers of solutions (by controlling important decisions and

transactions) without being labeled as social service users or social support receivers. This is important because the way needs are satisfied makes a big difference. Doing so is also important as it enables people not only to meet their needs but also to simultaneously create social capital and assets for further development (Elsen 2023).

The SSE and its role in providing social service and protection are very diverse and evident among various communities in different parts of the world. To provide examples, we discuss African, European, and transnational (in the case of immigrant communities) examples of SSE endeavors below.

### 3.2.3. African Examples of SSE

African SSE is ancient and based on African collectivist cultures that value self-help, mutual aid, and cooperativism. At the heart of this culture and African SSE are African philosophies such as Ubuntu (Hossein 2018). Ubuntu underlines the quality of being an authentic human (or the quality of human excellence) in terms of communal relationality at the individual, family, community, societal, environmental, and spiritual levels. Central to Ubuntu is relationality and caring about the well-being of others at all levels. In Ubuntu, caring for and supporting others is seen as a means of enhancing one's own well-being and authenticity. The maxim, "I am because we are," best summarizes the philosophy of Ubuntu (Mayaka and Truell 2021).[11] Such philosophies enable Africans to establish SSEOs and provide various forms of social services and protection to their communities. African SSEOs can be broadly classified into three main categories, i.e., cooperatives, sociocultural and recreational associations, and mutual benefit/aid societies (Fonteneau et al. 2011; Little 1962).

### Cooperatives

Various forms of formal and informal cooperatives are common in the SSE of various communities in Africa. These cooperatives can be producers', workers', consumers', or other types of cooperatives. These cooperatives operate in different sectors, including agriculture, health, finance, housing, and distribution. Examples of such cooperatives could be farmers' cooperatives in Africa, such as dairy farmers' cooperatives in Kenya. Such cooperatives have provided their members and communities with various opportunities and services including job opportunities, income security, worker protection, consumer protection, product and capital market, poverty reduction, education and training, networking, and social inclusion (Fonteneau et al. 2011).

### Sociocultural and Recreational Associations

Sociocultural and recreational associations of the African SSE can be formal or informal and include those organizations that primarily have social, cultural, and/or religious objectives. Examples of such associations include socioreligious associations, dance groups, drum groups, and various friendship groups found in different societies. Even though these associations are primarily designed for social and recreational purposes, their members regularly meet and contribute money and other resources (such as food) to fund their activities and help one another during various emergencies such as birth, death, and hospitalization. They provide for their members services and opportunities such as education, training, dialogues, recreation, initiation ceremonies, social assistance, physical labor support, social capital, and meaning to life (Fonteneau et al. 2011; Little 1962).

### Mutual Benefit/Aid Societies

Mutual benefit/aid societies of the SSE in Africa can be formal or informal and include, among others, burial societies, Rotating Saving and Credit Associations (ROSCAs), labor share groups (labor force rotation schemes), village grain banks, and occupational associations. These mutual benefit institutions and organizations function to provide various social protections and services for their members and communities. They do so by pooling resources and sharing various risks related to, among others, illness, accidents, death, old

age (retirement), unemployment, unexpected expenses, lack/loss of income or capital, and poor harvests (Fonteneau et al. 2011).

*Burial societies.* Burial societies in various African communities primarily serve as insurance by sharing risks and expenses in the case of death, illness, and accident. In doing so, they provide their members with various social services and protection by providing traditional psychosocial support to the bereaving family, helping orphans, assisting the elderly, helping the sick and disabled, assisting the poor and unemployed, and participating in community development activities (e.g., sanitation and environmental campaigns). An excellent example, in this case, is the Ethiopian burial society institution known as Iddir (see Box 1) (Aredo 2010; Pankhurst 2008; Pankhurst and Mariam 2000).

**Box 1.** Iddir: an Indigenous social insurance.

> *Iddir is one of the fundamental and enduring SSE institutions of Ethiopian society. Iddir is found in most parts of Ethiopia, including rural and urban areas. Many, if not most, Ethiopians participate in at least one Iddir.*
>
> *Iddir can be formed based on locality, gender, occupation, religion, and ethnic affiliation. It can be registered (formal) or unregistered (informal). Either way, it is well recognized by tradition, government, and civil society. Iddir is known to be broad-based, egalitarian, democratic, transparent, and accountable. Major decisions in Iddir are made in a participatory manner by majority vote during meetings involving all members. In Iddir, members come together and contribute a fixed amount of money regularly (e.g., monthly) to create money pools. They also contribute labor and other resources (e.g., food) on various occasions. They use such resources to address their members' and communities' death and death-related issues and to provide other social services. The main function of Iddir is providing various forms of support in times of bereavement. At the time of death, Iddir assists families that have lost loved ones. For example, Iddir provides a fixed compensation to a bereaved family. It also organizes and covers the cost of burial and related ceremonies. Furthermore, it provides traditional psychosocial support. In this case, for example, some members of the Iddir stay at the house of the bereaved family for a few days to help and console the family.*
>
> *In the case of other social services, Iddir helps members and their families when facing various social, medical, and financial problems. For instance, when members are unable to make contributions because of financial problems, the Iddir covers the cost of their contributions or exempts them from making contributions. In some cases, very poor members are also given chances to make a reduced contribution. Iddir also helps members during sickness, unemployment, old age and retirement, harvest loss and food shortage, property loss, and death of parents and breadwinners, by providing, for example, direct financial assistance or loans. Furthermore, Iddir provides support in the case of weddings and other celebrations by providing needed equipment and labor.*
>
> *Iddir has also become more and more involved in community development activities in Ethiopia. For example, it helps to mobilize members for various community development initiatives such as sanitation campaigns, environmental work, crime prevention activities, and campaigns against HIV/AIDS. In the case of HIV/AIDS prevention, for example, Iddir helps by providing education and training, information, statistics, and counseling. They engage in such activities by working together with the private, public, and civil society sectors. Iddir's role in strengthening and maintaining social bonds, social cohesion, and social capital is also indispensable (Aredo 2010; Pankhurst 2008; Pankhurst and Mariam 2000).*

*ROSCAs.* ROSCAs are enduring informal financial institutions found among various societies in different parts of the world. They are given different names in different societies. For example, they are called "Esusu" in Nigeria, "Chama" in Kenya, and "Equub" in Ethiopia. In ROSCAs, members make regular monetary contributions to a fund to be able to receive a lump sum in rotation based on the amount of their contributions. The basic principles of ROSCAs are regularity (regular contributions) and rotation (in receiving the fund). ROSCAs address risks that have mainly financial nature. However, ROSCAs have multifaceted financial, economic, and sociocultural functions. They enable the provision of savings (capital formation) and credit. In addition, they enhance and restore neighborhood and friendship solidarity. The services provided by ROSCAs enable members to strengthen their self-discipline for saving, receive timely and usually interest-free credit, purchase durable consumption goods, start or strengthen businesses and production, fulfill socio-cultural and psychological needs and obligations (e.g., helping families and friends), deal with various emergencies, and resist social and business exclusion. ROSCAs have several

advantages including accessibility (especially for those excluded or unbanked), flexibility, timeliness, adaptability, and multi-functionality (Ardener 1964; Aredo 2004; Bouman 1977; Hossein and Christabell 2022; Tadesse and Erdem 2023).

*Labor-share groups and village grain banks.* These institutions address risks related to mainly agricultural production (e.g., harvesting and weeding). In labor-share groups, members support one another by sharing their labor rotationally and reciprocally. The labor sharing could be in agriculture-related activities or other areas such as domestic work. Such activities enable members to maintain and strengthen their productivity and income sources, in addition to providing them with psychosocial support. Such groups are also known by different names in different societies. For example, in Ethiopia, they are called Debo in some areas and Wenfel in others (Habtu 2012).

In the case of village grain banks, the objective is to prevent food scarcity and starvation during difficult seasons. Village grain banks enable villagers to borrow food grain and survive difficult times. The borrowed food grain is to be paid back at the next harvest. Government agencies and NGOs support such community-owned grain banks in their effort to address food shortage and starvation and enable food security and food self-sufficiency (Yameogo 1997).

*Informal occupational associations.* These associations are set up to address the risks associated with a particular occupation. In such associations, people of the same occupation come together and help one another. These associations could be established and run by, for example, market women (e.g., foodstuff sellers, sugar sellers), carpenters, shoemakers, drivers, seamen, cooks, and stewards. The services provided by such associations could help to discourage competition, reduce the cost of inputs, protect the status of their occupation and members, and promote the remuneration of their members (Little 1962).

In general, the above SSEOs and institutions in Africa are mainly characterized by cooperativism, self-organization, self-financing, self-help, and mutual aid. Such characteristics are what enabled them to play a key role in providing available, accessible, and affordable social protection measures and social services for Africans in various societies. They provide not only contributary forms of protection such as social insurance but also non-contributary ones such as social assistance. Furthermore, they provide their members and communities with social care services (e.g., in case of exclusion, abuse, loss of parents, etc.) and labor market services (e.g., creating employment and maintaining employment standards). Such social protection measures and services are crucial given the limited formal means of social protection and services (provided by the public and private sectors) in many communities of Africans. Regardless, such institutions and organizations have proved to be enduring and robust; they have stood the test of time and are likely to continue to exist in the future. This is so because of their multifunctional nature and advantages that involve psychological, sociocultural, economic, and political dimensions. Many of these institutions and organizations, although they are autonomous, also showed capacity in the co-creation of social protection measures and social services with the public and private sectors.

### 3.2.4. European Example: Italian Cooperatives

In Italy, following the economic crises and social changes of the 1970s, cooperatives emerged as synergetic and creative solutions to societal problems, compensating for the shortage of public solutions for social needs such as childcare, eldercare, care for people with disabilities, labor integration of disadvantaged and disabled people, and re-integration to society of drug users (Elsen 2023).

Cooperatives in Italy are fostered by legal frameworks, public administration, regional and national consortia, and a mutuality fund (Elsen 2023). There are various legal forms of cooperatives identified in the country's civil code. These include agricultural cooperatives, worker cooperatives, service cooperatives, and social cooperatives. For the purposes of this paper, however, we only discuss social cooperatives (Bianchi 2021).

Italy introduced a legal framework for social cooperatives in 1991. According to the law, the purpose of social cooperatives is "pursuing the general interest of the community

in human promotion and social integration of citizens" (Bianchi 2021). The law also makes a distinction between type A social cooperatives that provide social and healthcare services and type B social cooperatives that emphasize the qualification and employment of disadvantaged groups such as immigrants. Such cooperatives, to achieve mandates that mainly have to do with social inclusion, provide educational, healthcare, and social services. In addition, they engage in socioeconomic activities in different sectors of the economy. They do so by following democratic, integrative, and participative rules (Elsen 2023).

An interesting specific example, in the housing sector, is Haus der Solidarität (HdS) (in German) or Casa della Solidarietà (CdS) (in Italian). This is a type A social cooperative that is based on the SHM (Hull-House tradition) concept. The cooperative has been providing much needed housing and other social services for immigrants and other residents in a three-story building in the town of Bressanone/Brixen in Northern Italy for more than, 20 years.[12] It is also preparing to open up new accommodation in a nearby small town.

Social cooperatives are also common in the agriculture sector, especially in what is known as social agriculture, which combines agricultural activities with social and healthcare objectives. In this case, many social cooperatives are associated with community-supported agriculture initiatives and solidarity purchase groups to sell their products (Elsen 2023; Lintner and Elsen 2020). An excellent example, in this case, is Barikama, a type B social cooperative, which was established by PAD in Rome (Lintner and Elsen 2020). We will discuss Barikam in the next section.

### 3.2.5. Transnational Example: PAD and Their SSEOs in Europe

PAD are found in many different countries and regions of the world, outside of Africa. For example, 15 to 20 million PAD live in Europe (Congress.gov 2019). According to the UN Office of the High Commissioner for Human Rights (OHCHR n.d.), "Whether descendants of those Africans that were displaced to the Americas during the transatlantic slave trade many generations back, or more recent migrants who have journeyed to the Americas, Europe, Asia and within Africa itself, people of African descent throughout the world make up some of the most marginalised groups". Studies by international and national bodies show that PAD still have limited access to, among others, quality education, health services, housing, and social security (UN n.d.).[13] They are also victims of social and business exclusion (Hossein 2018).

However, PAD do not remain passive when facing various forms of social and business exclusion. They self-organize and engage in the SSE of PAD to cope with and fight against such problems (Hossein 2013).[14] The SSE plays a crucial role in the lives of PAD. The SSE in general and the SSE of PAD, in particular, is where PAD find refuge and meet their economic and social needs. It is also where they contribute to the economy and society (Hossein 2017).

Types of SSEOs of PAD

Our scoping review has identified 20 types of SSEOs of 18 different PAD communities (e.g., Ghanaians, Ethiopians, African Caribbeans, etc.) in 16 different European countries (e.g., the UK, Germany, the Netherlands, etc.). The types of SSEOs of PAD include: hometown associations, development associations, social movement-oriented organizations, women's organizations, cultural associations, umbrella organizations, cooperatives, housing associations, ROSCAs, Accumulating Saving and Credit Associations (ASCAs), burial societies, urban commons, social enterprises, professional associations, student associations, online-based associations, foundations, SSE programs and projects, local sports clubs, and other migrant organizations with social and economic objectives.

SSEOs of PAD are diverse and they can be many in number in a given city or country. For example, a study by Lampert (Lampert 2009) identified 325 different Nigerian organizations in London alone. Another study in Germany identified 140 Nigerian organizations (Marchand et al. 2015). In Denmark, a survey included 123 African diaspora organizations representing 22 countries and 3 regions in Africa (Trans and Vammen 2011).

Activities of SSEOs of PAD

SSEOs of PAD engage in a multitude of activities and services that can be classified into 11 categories. The first is education, training, and knowledge production activities and services (carried out in both the host and origin countries). These include early childhood education, primary and secondary schooling; before and after school services; language lessons; mentoring; entrepreneurship training; seminars, conferences, commissions, panels; consciousness-raising training; research work and publication; and newspapers, magazines, book series, and multimedia projects.

The second pertains to development activities and projects (especially for origin countries). These include the provision of financial, material, and social remittance; construction of schools, libraries, hospitals, wells, community halls, roads, etc.; promotion of environmental protection and neighborhood cleaning activities; and promotion of fair trade (e.g., between host and origin countries).

The third concerns social and cultural activities (usually carried out in the host countries) in which SSEOs of PAD organize and participate in different social and cultural activities such as arts, recreation, and religious activities. Specific examples, in this case, include cultural galas, life event celebrations (e.g., weddings), sports competitions, beauty contests, fashion shows, annual festivals, annual dinners, dance parties, religious and cultural holiday celebrations, food bazaars, art exhibitions, and participation in international carnivals representing origin countries.

The fourth is the provision of welfare and other social services (carried out in both the origin and host countries), including organizing daycare; helping victims of abuse; responding to urgent calls for help, emergency assistance (distribution of food, material goods, and money); supporting asylum seekers and other migrants; supporting senior citizens; supporting single-family households; supporting orphans; and supporting job seekers.

The fifth is about advocacy, activism, lobbying, and engagement in social movements for visibility, recognition, acceptance, rights, and social change (in both host and origin countries). In this case, efforts were made, for instance, to represent the interests of PAD, influence public and social policy decision making, and support PAD political candidates.

The sixth has to do with financial services carried out mainly by ROSCAs, ASCAs, and burial societies (in host countries). The services include savings, credit, and insurance (e.g., in case of illness and death).

The seventh concerns housing services (in host countries). For instance, in the 1980s, members of a Black housing federation in the UK provided homes for 33,000 households (Gardiner 2006).

The eighth is agricultural (e.g., farming and gardening) activities (conducted in host countries) that produced food (e.g., vegetables and dairy products) for consumption and market. An excellent example in this case is Barikama (see Box 2 below).

**Box 2.** Barikama: A Social Cooperative.

*Barikama is a social cooperative in the Italian agriculture sector. It was established in 2011 near Rome by six African immigrants who had experienced severe labor exploitation and racial discrimination in the Italian agriculture sector in Rosarno in the region of Calabria. The cooperators' origin countries included Mali, Senegal, Guinea, Benin, and Gambia. The term Barikama, which comes from the Bambara language of Mali, means "resilience" and signifies adaptation to change, overcoming traumatic events, and transforming difficulties into opportunities. The main function of Barikama is the social and employment integration of members and the local community through the production and sale of organic agricultural products (e.g., yogurt and vegetables) at local markets. By doing so, cooperators were able to create decent jobs and generate income for themselves; create internship and employment opportunities for young people (especially those with autism) in the local community; increase their social contact and interaction with members of the local community; improve their Italian language skills much more quickly; focus on environmentally friendly production and delivery using bicycles; and collaborate with local solidarity purchase groups (Barikama n.d.; Barikama 2023. Message to authors, February 08; Siad 2015).*

The ninth encompasses social business enterprises and activities (in host countries). For example, in the 1980s and 1990s, the Nation of Islam (NOI) had restaurants, shops, grocery stores, bookstores, clothing stores, and gardens in the UK (Tinaz 2006).

The tenth pertains to a consultative and networking role (in both host and origin countries). This includes bridge-building services between PAD communities and host and origin countries, as well as networking with similar PAD organizations, other SSEOs, professional groups, and the public sector.

The final is health services, addressing physical and psychological health (in both host and origin countries). Specific activities, in this case, include the distribution of medicine, family planning services, HIV/AIDS prevention, female genital-cutting prevention, and treatment of major diseases (e.g., cancer, eye-related problems) with the help of specialists from among community members and other partners.

Contribution of SSE of PAD

SSEOs of PAD in Europe make multifaceted contributions to PAD, host countries, and origin countries. Their contribution can be classified into five categories. The first is helping to meet the fundamental needs of PAD in Europe, for example, by filling gaps created by the private, public, and "mainstream" third sectors. The various needs addressed in this case are economic (e.g., financial, material, employment, food), sociocultural (e.g., conviviality, friendship, networking, sense of belonging, collectivism), psychological (e.g., self-esteem, happiness, fun), physical health (e.g., culturally appropriate health services), and spiritual (e.g., traditional spiritual practices, mosque- and church-related practices).

The second concerns promoting social inclusion (integration) as a two-way process. In this case, SSEOs of PAD help to maintain the cultures and identities of, as well as contacts and ties with, the origin countries. They also help in the struggle for recognition and visibility as PAD, particular groups of PAD, citizens of the host countries, or significant minority groups in the host country. At the same time, they facilitate participating in and learning from the culture of the host society by, for example, creating opportunities for learning local languages, increasing contacts and interaction with people of the host country, and engaging in housing, job, and financial markets. They take part in cultural exchanges and appreciation of diversity. In so doing, they significantly contribute to local, subnational, national, and regional social inclusion policies and initiatives.

The third is about fostering the empowerment of PAD at the individual, family, group, and community levels. This is achieved, for example, by enabling self-help and self-reliance of PAD (e.g., PAD women having control over their financial resources as a result of ROSCA participation); raising the consciousness and dignity of PAD; and facilitating PAD's transition from being transient migrant communities (asylum seekers and refugees) to settled immigrant communities.

The fourth concerns supporting social development, especially in origin countries. They do so in two main ways: (a) by enabling various forms of remittance, i.e., financial, material, and social remittance; and (b) by directly constructing schools, hospitals, roads, wells, etc. Their intimate knowledge of local conditions and needs in the origin countries helps them to be more effective than other development organizations (e.g., international NGOs) at promoting social development.

Finally, the fifth contribution has to do with diversifying economic activities in the host countries and Europe. They do so, for example, by introducing new or alternative ways of doing business and embedding economic activities into social and cultural contexts. An excellent example, in this case, is ROSCA, which is a non-capitalist form of finance that balances social and financial/economic objectives in an excellent manner.

## 4. The Parallel between the SHM and SSE

From the foregoing discussions, it can be seen that there are substantial parallels between the SHM, especially that of the Hull-House tradition, and the SSE, especially SSE by vulnerable groups (self-organized SSEOs) and SSE with vulnerable groups (collaborative SSE efforts).[15]

Before we start discussing the parallels, let us first mention two main points of differences between the two. The first difference is that the activities of Hull-House occurred in a historical time when, despite the presence of a myriad of social problems, there were limited or no social rights, policies, and infrastructures. In contrast, current SSE activities are happening in a time when there are social rights, policies, and infrastructures (even though such provisions are inadequate, and in many cases implemented discriminatorily, for example, based on immigration status and citizenship). The second point of difference, especially comparing Hull-House with the SSE of PAD, is that while most Hull-House initiatives were "supported self-help" (i.e., SSE with vulnerable groups where the initial initiatives came from Addams and her colleagues),[16] most of the SSEOs of PAD were self-initiated and organized by the immigrants themselves.

In the case of the parallels, four major interrelated points of similarity between the two can be identified. These are as follows: the non-positivist and non-individualistic philosophical base; anti-capitalist nature; special experience and contribution during times of crises such as the "immigration crises;" and innovative methods of addressing social problems.

### 4.1. Non-Positivist and Non-Individualistic Philosophical Base

The first parallel between the two is related to their non-positivist and non-individualistic ontological and epistemological stances, which is in contrast with the stances of capitalism and the COSs tradition of SW. For example, we have seen that Addams herself was one of the main pragmatist philosophers and advocates of collectivism/cooperativism. Hence, Hull-House's responses to social problems was informed by pragmatism that located the main cause of social problems in the social structure and underscored collective action and social reforms (Morrison 2016).

A similar ontological worldview can also be noted in the case of the SSE in general and the SSE of Africans and PAD in particular. The SSE is based on a pre-analytic vision of the economy as an open and interdependent subsystem of the ecosystem/environment (as opposed to a closed and disembedded system). The SSE assumes that the economy is a social construct, which can be shaped and reshaped to help meet fundamental human needs including ecological, social, and economic needs (Elsen 2018). Furthermore, it assumes that humans are relational or whole beings who are rational, emotional, and normative, instead of being only rational. It also asserts that economic agents are not autonomous but socially situated. SSE advocates for the superior ontology of the embedded agency. Consequently, the SSE is based on a cooperative logic (emphasizing cooperation and reciprocity, morality and common good) as opposed to a competitive logic (Dash 2016). The SSE of Africans and PAD is also based on assumptions, values, and principles derived from African collectivist philosophies such as Ubuntu that enable the embedding of economic activities into sociocultural, environmental, political, and spiritual contexts.

In general, "the SSE is conceptually anchored on a 'flourishing services' approach, or a social provisioning approach to economics. It may as well be referred to as the 'sustainable livelihood' approach, which refers to the study of how humans make a living from their social and natural environment in a socially just, environmentally responsible, and morally correct way." SSE provides us with a more realist, non-essentialist social ontology (Dash 2016, p. 10).

### 4.2. Anti-Capitalist Stance and Response to the Capitalist Hegemony and the Resultant Crises

The second parallel is that both the Hull-House tradition and SSE are anti-capitalist and anti-laissez-faire. They recognize that the capitalist system is the main cause of the multiple crises the world is experiencing. Accordingly, they respond to the capitalist hegemony and its multiple crises by proactively providing time-tested as well as new experimental solutions. They provide alternative ways of producing goods, services, and knowledge while displaying solidarity.

SW itself was created as a response to the multitude of social problems resulting from rapid industrialization and urbanization. Unlike advocates of capitalism, Addams and the

SHM tradition of SW located the main cause of individuals' poverty in the macrosystem or in the social conditions. Accordingly, they promoted collective and democratic group and community-based SW as well as social action and reform for, for instance, improved working and living conditions in industrial cities (Addams 2008; Eberhart 2009; Franklin 1986; Melgar et al. 2021; On the Commons 2010). Addams strongly believed in the power of cooperation over competition. She was also deeply influenced by Beatrice Webb's work *The Co-operative Movement in Great Britain*. "To Addams, cooperation was not only an alternative approach to organizing economic activity, but also a principle of social democracy" (On the Commons 2010). Addams and Hull-House's work was not limited to supporting activities that are against the exploitative economy and repairing the social and ecological destruction it causes. Their work had a controlling, correcting, and limiting effect in the economic and political sphere. It developed independent social economies and enforced comprehensive social participation against dominant capital interests (Elsen 2011).

Similarly, most current SSE initiatives in industrialized countries are responses to the inhumane nature of the capitalist hegemony. Time and time again, we have observed that SSE is a rescuer (or a place where everybody looks for solutions) during times of crisis such as the great depression and the 2008 financial crisis. We have also noted in our literature review that the SSE of PAD in industrialized countries exists mainly because of PAD's and Africans' unmet needs and the social and economic exclusion that are created and exacerbated by the capitalist hegemony. SSE aspires to negate the capitalist hegemony and the resultant multiple crises by working to re-embed economic activities into social, cultural, and ecological contexts and make the economy controlled by local people so that the economy will be life-serving instead of life-threatening (Elsen 2018).

### 4.3. Experience with the "Immigration Crises"

Among the main crises created and exacerbated by the capitalist hegemony are the so-called "immigration crises." Both the Hull-House tradition of SW and current SSE have unique experience in addressing the immigration crises of their particular times. As discussed above, professional SW originated in the late 19th century in the USA as a response to social problems especially affecting those tens of millions of European immigrants associated with the mass migration "crises" (Franklin 1986; NASW n.d.b). This situation was what Addams and her colleagues wanted to address by establishing the Hull-House in impoverished immigrant neighborhoods where they lived and worked with the immigrants.

Unfortunately, "migration crises" are still happening at this stage of "civilization." A good example, in this case, is Europe's "migration crisis" of 2015, in which more than 1 million migrants arrived in Europe (Greenhill 2016). This time was full of tragedies involving loss of life, human rights violations, racism and social exclusion, a shortage of social services such as education, etc. (BBC 2020; Spindler 2015). The situation also required the intervention of SSE. For example, many SSE initiatives (whether self-organized or not) in Europe are targeted at improving the inclusion and well-being of immigrants by providing various social services. The same is true in the case of the SSEOs of PAD in Europe.

In short, in both periods, we observe that when the global capitalist hegemony (including the public and private sectors) fails vulnerable immigrants, both the Hull-House tradition of SW and SSE were there to provide innovative solutions and address the social problems affecting immigrants.

### 4.4. Innovative and Realistic Solutions to Social Problems

The innovative solutions of the Hull-House tradition and the SSE are based on group and community work that emphasize collective and social action. "Addams introduced the notion that a well-organized community life and culture can exist even among poor people and that one means by which to make society better is to attend to and nurture that community life and culture" (Specht and Courtney 1995, chp. 1). Hull-House itself was a cooperative undertaking that included within it various SSE initiatives and organizations. In addition to organizing SSEOs and initiatives, Hull-House carried out other

community-based activities contributing to SW research and education, social movements, and social reforms.

Hull-House's work was guided by three basic principles, i.e., respect for the dignity of all people, participation of and collaborative work with community residents, and the conception of the causes of poverty and inequality as a structural phenomenon (Malekoff and Papell 2012). Hull-House's work can be summarized in the motto of the three Rs, i.e., Residence, Research, and Reform (Branco 2016).

Similar approaches are observed in SSE. For example, in the case of SSE in general, the following innovative aspects are noted: bottom-linked governance and democratic decision-making; participation, self-determination, and empowerment of users; integrative and cooperative knowledge production (as opposed to the concentration of knowledge and dependencies on professionals); sector-transgressing solutions that combine resources and adapt to specific needs; the mix of material and nonmaterial resources from different sources; and integration of collaborating social networks, volunteers, and stakeholders (Elsen 2023). In the case of SSE in Africa and SSE of PAD in Europe, it is noticed that their institutions and organizations are founded on the principles of self-organization and cooperativism, enabling them to provide available, accessible, and affordable social protection and services.

In both Hull-House and SSE cases, the secret to their innovative and realistic solutions to social problems lies in how the fundamental needs of people are met. In both cases, service users' needs were met while they were seen as participants, co-creators, and people with dignity and agency. For instance, Addams and her colleagues believed that being a service user should not be a degrading experience (NASW Foundation n.d.).

To understand the significance of such approaches, Max-Neef's (2017) concept of fundamental human needs is indispensable. The key idea in his work is that the way we satisfy fundamental needs can make a great difference since not all satisfiers are equal. There are synergetic satisfiers, which are capable of satisfying multiple needs at the same time. For example, infant formula and breastfeeding are both satisfiers of the need for subsistence. However, breastfeeding is a much better satisfier since it simultaneously satisfies other needs such as needs for protection, affection, understanding, participation, identity, and freedom (Alliance for Sustainability n.d.; Max-Neef 2017). Similarly, consuming vegetables bought as economic goods and consuming vegetables produced in a social cooperative have different qualities in terms of their contribution to empowerment, social inclusion, well-being, and freedom from market dependencies (Elsen 2023).

## 5. Conclusions and Recommendations

In this paper, we have highlighted SW's major inconsistencies that have hindered the profession from realizing its full potential in the 21st century. These inconsistencies are in relation to SW's (i) ontology, epistemology, and methodology; and (ii) mutual dependency on the capitalist hegemony, which is the main cause of the multiple crises that primarily affect vulnerable groups.

We have argued that SW needs to address its inconsistencies and transform itself to remain relevant in the rapidly changing world. We have proposed that SW should look back and look around to move forward. By looking back, SW can learn from the SHM tradition, epitomized by the works of Jane Addams and her colleagues at Hull-House, which emphasized group and community-based social work as well as social action and reform. By looking around, SW can learn from the SSE, which re-embeds economic activities into sociocultural and ecological contexts and makes the economy life-serving instead of life-threatening.

Our discussion of the SHM tradition and the SSE shows that learning from these two similar approaches can help SW to address its inconsistencies and realize its contemporary mission of promoting social change and development, social cohesion, and the empowerment and liberation of people. Doing so enables SW to truly be the profession that specializes in social sustainability.

We are not arguing here for abandoning the individualistic approach in SW altogether. Given the variety and complexity of problems social work addresses, it is important for the profession to be versatile and have a complete toolbox of strategies and interventions, including the individualistic approach. What we are asserting is that SW's dominant ontology, epistemology, and methodology should be based on a collectivist, cooperativist, and community-based tradition while acknowledging the importance of the individualistic approach.

We recommend that SW should emphasize SSE and the SHM legacy by (i) teaching its students and practitioners about SHM and the Hull-House tradition of SW as well as current concepts and practices of SSE; (ii) establishing SSEOs and projects; (iii) referring service users to SSEOs and projects for employment, internship, volunteering, and visits; (iv) encouraging service users to establish SSEOs; (v) providing training for members of SSEOs; (vi) collaborating and cocreating with SSEOs; and (vii) advocating for SSEOs to help them address major challenges such as financial difficulties, lack of visibility, and lack of recognition.

We hope that the discussions and examples provided in this paper initiate further discussions and analyses on the topic and serve as a reference point for those in the fields of SSE and SW, especially for SW students, practitioners, researchers, and educators.

We conclude by emphasizing that SW has a crucial role to play in addressing the multiple crises of our time, especially in relation to social sustainability, but it needs to address its fundamental inconsistencies.

**Author Contributions:** Conceptualization, M.E.T. and S.E.; Methodology, M.E.T.; Validation, S.E.; Formal analysis, M.E.T.; Investigation, M.E.T.; Resources, S.E.; Writing—original draft preparation, M.E.T.; Writing—review & editing, M.E.T. and S.E.; Visualization, M.E.T.; Supervision, S.E.; Project administration, S.E.; Funding acquisition, S.E. All authors have read and agreed to the published version of the manuscript.

**Funding:** This article is a product of the ASTRA research project (https://www.jyu.fi/en/research/astra, accessed on 9 December 2022) which has received funding from the European Union's Horizon 2020 research and innovation programme under the Marie Skłodowska-Curie grant agreement No 955518. The APC was funded by the same project. The article reflects only the authors' views and the European Commission is not responsible for any use that may be made of the information it contains.

**Institutional Review Board Statement:** Not applicable.

**Informed Consent Statement:** Not applicable.

**Data Availability Statement:** Data sharing is not applicable to this article.

**Acknowledgments:** We are grateful to the editors, the four anonymous reviewers, and Walter August Lorenz for their insightful feedback on this paper. We also appreciate the support of our colleagues and partners in the ASTRA project.

**Conflicts of Interest:** The authors declare no conflict of interest.

## Notes

[1] It is important to underline that the boundaries of what counts as SW have always been fluid and contested. Nevertheless, this paper is about the profession of SW, which, like other professions, has its own formalized education and training, code of ethics, and autonomy (self-regulation).

[2] Our review followed five key steps (i) identifying specific research questions; (ii) identifying relevant studies; (iii) selecting studies (with a two-stage screening process); (iv) charting data from selected studies; and (v) collating, summarizing, and reporting the results (Arksey and O'Malley 2005).

[3] It is also important to acknowledge that these two traditions complemented (exchanged with and enriched) each other in some ways (Thompson et al. 2019).

[4] Note that there was not a straight transition or connection between the psychiatric/psychotherapy-based SW and managerialism in SW. In the 1970s, for example, there was a strong movement towards radical social work, which was opposed to case work and had commonalities with SHM. However, this movement started to decline in the early 1980s (Ferguson 2009). Furthermore, it is

important to note that, while psychological/psychiatric-based SW developed out of SW itself, managerialism was imposed on SW at a political level (Lorenz, Walter 2022. Message to authors, December 03).

5    Addams and Richmond are not the only pioneers of SW. Different communities and countries have their own pioneers who were contemporaries of Addams and Richmond. These include, for example, Black, Indigenous, and other People of Color (BIPOC) SW pioneers such as Eugene Kinckle Jones, W.E.B. Dubois, and Frederick Douglass (See Wright et al. 2021), as well as other pioneers from Europe such as Alice Salomon of Germany.

6    Toynbee Hall was founded in 1884 in the East End of London, by the married couple Samuel and Henrietta Barnett. Toynbee Hall introduced a new way of addressing social problems to achieve social change. Their approach was different from the then conventional individualized and piecemeal approaches. They pioneered a social settlement where future leaders, students, and scholars (of Oxford and Cambridge Universities) lived and worked as volunteers with the poor in an impoverished industrial area. Some of their contributions include: participation in local life; development of adult education; collection of social data; and improvement of local social and industrial conditions (Encyclopaedia Britannica n.d.b; Toynbee Hall 2018).

7    Settlement workers were middle-class and affluent volunteers who "settled" in the immigrant districts of large cities (Stuart 2019). The residents paid for housing and food. The majority of them had jobs outside of Hull-House. However, their work for Hull-House was voluntary (Addams 2008; Eberhart 2009).

8    Here it is important to acknowledge that Addams is criticized for not making racial justice the central focus of her work and for doing little to support, for instance, African Americans and Native Americans (The Jane Addams Peace Association n.d.; Lynn 2018).

9    Hull-House was known for its efforts to hand over to the public authorities activities it initiated, as long as the public authorities were able run them properly (Addams 2008).

10    "A cooperative is an autonomous association of persons united voluntarily to meet their common economic, social and cultural needs and aspirations through a jointly-owned and democratically-controlled enterprise. . . . Cooperatives are based on the values of self-help, self-responsibility, democracy, equality, equity, and solidarity" (ICA n.d.).

11    See Tusasiirwe et al. (2021) for discussions on the challenges of Ubuntu.

12    At the time of writing this paper, the first author was living in the HdS. The second author also contributes to the cooperative financially and as a mentor.

13    For further discussion of the condition of PAD and other immigrant groups in Europe, see De Genova (2018).

14    For discussions on SSE of PAD in the Americas, see works by Hossein (Hossein 2018) and Nembhard (Nembhard 2014).

15    In this case, a third type of SSE could be SSE for vulnerable groups (welfare- or handout-based SSE).

16    It is important to remember that the clubs and cooperatives at Hull-House were self-governing (On the Commons 2010).

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
