# Peer review of "The Social Solidarity Economy and the Hull-House Tradition of Social Work: Keys for Unlocking the Potential of Social Work for Sustainable Social Development"

_socsci, doi:10.3390/socsci12030189_

Round 1
Reviewer 1 Report
Interesting and worthwhile contribution to history of ideas in social work. This particular historical period is often overlooked or simply skimmed over. English language use needs further work.
Reviewer 2 Report
I really liked this paper. While not new, the (divergent) historical roots of social work with Mary Richmond and Jane Addams bear repeating for new generations of social workers and the author does this well. These earlier historical divergences assume a particular importance in contemporary social work where the dominant (and often unthinking) push is to turn more and more to psychology for answers (and methods through which to approach) social problems.
I was glad to see the author also identify some of the fundamental gaps between ontology, epistemology and methodology. While social work has undergone something of an ethical turn in recent decades, it is only recently that I have become aware of it looking to other areas of philosophy such as ontology and epistemology and I am convinced of the need for some greater alignment between what social work is and the knowledge base and methods that it draws upon. In this sense Jane Addams' connection to Dewey's pragmatism seems t make sense - it offers the potential to reframe social work as a broadly socio-educational (or socio-pedagogical) endeavour rather than a treatment or therapeutic one.
I am less well-versed in ideas of SSE but I do think there is a need for a fundamental re-thinking of the role social work fulfils in society and some of the more collectivist examples provided offer very different possibilities for understanding and doing social work. It potentially takes the focus of social work beyond the individual (Western) paradigms that currently frame it. The need to introduce alternative economic models is well-made.
The author explains how he/she went about the knowledge acquisition for the article and this adds a layer of rigour. Writing and argumentation, generally, is very good.
A couple of small points that the author might want to consider: 1) On page five, the author seems to move very quickly from the early 20th century to managerialism. There was quite a bit in between - some of it (in the UK at least) pretty progressive, such as movements towards radical or community social work. In the other hand, social work has become co-opted within a statist paradigm and has become caught up in ideas such as child protection, which can be made out to be progressive when in most cases they reinforce the individualistic response to social problems. But, such ideas of protection will be difficult to move on.
The author points to the African idea of Ubuntu. I have been concerned for some time now that this can be adopted in a somewhat romanticised way. I think the situation in Africa regarding social relations is rather more complicated. So, while qualities of collectivism and interdependence might be aspired to, there is nor ready made way to their realisation.
I agree with the author in the Conclusion that social work would do well to look back to look forward and I think an article like this offers an important contribution to its capacity to do so.
Reviewer 3 Report
This paper argues that the theory and practice of social work can be substantially informed by knowledge about 1) the settlement-house movement pioneered by Jane Addams at Hull House, and 2) the social solidarity economy of people of African descent in various parts of the world, most notably, Europe. The author analyzes a large body of research on the settlement house movement and the social solidarity economy – for instance, the author conducts a “scoping review” of over 11,000 studies of the social solidarity economy. The paper concludes that the analysis fills knowledge gaps and thereby advances social work theory and practice.
On the positive side, the paper identifies a potentially interesting problem in social work theory and practice (lines 21-92), and it seems to provide a comprehensive bibliography (lines 937-1120).
On the negative side, however, there are several problems that undermine the paper’s potential for contributing to the theory and practice of social work.
First, the description of the methods (lines 93-113) leaves out some important details. The “scoping review” method (lines 108-110) should be described in more detail. What are its advantages and disadvantages? What published studies have used this method (citing examples)? Moreover, what does the author mean by “screened” and “fully reviewed” (line 111)?
Second, the paper is far too long and covers material that is well-known to scholars who study social service delivery and the social and economic adjustment of immigrants. No new information or insights are presented in the sections on the settlement-house movement (lines 115-415) and the social solidarity economy of people of African descent (lines 416-764). These sections strike me as literature reviews that would appear in a thesis or a textbook rather than as novel analyses that would be published in a peer-reviewed journal.
Third, the section that draws parallels between the settlement-house movement and the social solidarity economy (lines 765-909) is not very compelling. The author draws parallels that are very general and revolve around the collectivist orientation shared by the settlement-house movement and social solidarity economy. The author also makes assertions that are based more on ideological views than on empirical evidence – for example, the claim about “capitalist hegemony” being the “main cause” of the world’s problems (lines 817-818).
Fourth, the short concluding section (lines 910-936) is very thin, especially after 20 single-spaced pages of text. This section, with all due respect, falls flat.
In closing, this paper is really two manuscripts in one – a manuscript on the settlement-house movement, and a manuscript on the social solidarity economy of people of African descent. The author’s attempt to join the two together in the paper’s last two sections does not produce a substantial contribution to social work theory and practice, especially since most of the material is already covered in textbooks and other readings that are assigned to social work students.
Reviewer 4 Report
1. Social work in fact has two directions—"social casework” and “settlement house movement” (SHM), instead of abandoning the SHM. In fact, the former direction has long been considered ineffective and replaced by current “professionalism” (medical-oriented though). The SHM has been evolved into community work in U.S. In both such new directions, professional social workers are involved. Community work focuses on social or structural changes, but these changes usually take long time and there are also always new social problems. In other words, individual difficulties/problems during struggles of social/structural changes continue and they need individual intervention. Of course, it is important not to blindly blame on individuals for their difficulties/problems. In other words, social work profession addresses both individual and societal needs. I personally have been a licensed social worker in therapeutic services as well as community organizing work.
2. SHM apparently sown the seed of social changes, fostering the rise and development of SSEs. SSEs do not need social workers because their social consciousness and self-governance.
Round 2
Reviewer 1 Report
Requires some changes to construction of argument and English language use.
Author Response
Thank you very much again for the invaluable feedback; the comments and suggestions have helped us improve our paper! We have made the following modifications:
- The English language use has been improved with the help of a professional English language editing service (i.e., MDPI English editing service). We have also thoroughly read the paper and improved its readability in all sections.
- We have improved the articulation and consistency of our arguments by editing the abstract, introduction, methods, results and discussion, and conclusion sections. For example, we have improved the flow of the methods section. We have also improved the section ‘the concept and practice of SSE’ by making it concise and by editing the numbering of its subsections. Furthermore, we have rewritten and improved the conclusion section of the paper.
Reviewer 3 Report
I appreciate the authors' efforts, but the changes do not adequately address the fundamental problems that I noted in my earlier report.
Author Response
Thank you very much for the comments and suggestions; they have helped us improve our paper. We have made the following modifications.
We have used a professional English language editing service (i.e., MDPI English editing service) to improve the language use and readability of the paper. We have also thoroughly read the paper and improved its readability in all sections.
We have edited the abstract, introduction, methods, results and discussion, and conclusion sections to improve the articulation and consistency of our arguments. For example, we have improved the section ‘the concept and practice of SSE’ by making it more concise and by editing the numbering of its subsections. We have also rewritten and improved the conclusion section of the paper.
Finally, as we indicated in our previous response, we would like to acknowledge that we respectfully disagree with two comments (point 2 and point 3) of the reviewer's earlier report.